# Synthetic Study toward Triterpenes from the *Schisandraceae* Family of Natural Products

**DOI:** 10.3390/molecules28114468

**Published:** 2023-05-31

**Authors:** Pavle Kravljanac, Edward A. Anderson

**Affiliations:** Chemistry Research Laboratory, Department of Chemistry, University of Oxford, 12 Mansfield Road, Oxford OX1 3TA, UK

**Keywords:** natural products, total synthesis, ynamides, cascade reaction, electrocyclisation, Suzuki coupling, medium ring synthesis

## Abstract

Triterpenoid natural products from the *Schisandraceae* family have long presented a significant synthetic challenge. Lancifodilactone I, a member of the family not previously synthesized, was identified as a key natural product target, from which many other members could be synthesized. We envisaged that the core ring system of lancifodilactone I could be accessed by a strategy involving palladium-catalysed cascade cyclisation of a bromoenynamide, via carbopalladation, Suzuki coupling and 8π-electrocyclisation, to synthesize the core 7,8-fused ring system. Exploration of this strategy on model systems resulted in efficient syntheses of 5,6- and 5,8-fused systems in high yields, which represent the first such cyclisation where the ynamide nitrogen atom is ‘external’ to the forming ring system. The enamide functionality resident in the cascade cyclisation product was found to be less nucleophilic than the accompanying tri-/tetrasubstituted alkene(s), enabling regioselective oxidations. Application of this strategy to 7,6-, and 7,8-fused systems, and ultimately the ‘real’ substrate, was ultimately thwarted by the difficulty of 7-membered ring closure, leading to side product formation. Nevertheless, a tandem bromoenynamide carbopalladation, Suzuki coupling and 6/8π-electrocyclisation was shown to be a highly efficient tactic for the formation of bicyclic enamides, which may find applications in other synthetic contexts.

## 1. Introduction

Triterpenes from *Schisandraceae* species have attracted attention from the synthetic community due to their intriguing molecular frameworks and biological activity [1,2]. Efforts by the groups of Yang [3,4,5,6,7], Li [8,9], Tang [10,11] and Anderson [12,13] resulted in 11 total syntheses so far. However, previously reported strategies often suffer from lengthy synthetic routes and challenges in applications to other members of the family. We aimed to develop an efficient approach to the *Schisandra* triterpenes in which lancifodilactone I (**1**, Figure 1a), a previously unconquered target, was identified as a potential common precursor to other natural products in this family. We envisioned that the key 7,8-fused ring core of lancifodilactone I could be synthesized via a palladium catalysed cascade reaction: starting from bromoene-ynamide **2**, oxidative addition followed by carbopalladation (**3**), Suzuki coupling (with a suitable dienyl organometallic) to give **4**, and 8π electrocyclisation would deliver the 7,8-fused system **5** in one step. An electron-donating ynamide functionality in **2** was proposed to enhance the nucleophilicity of the resulting enamide in **5** for further functionalisation, while the bulkiness of the ynamide nitrogen substituent should prevent *cis-trans* isomerisation of the intermediate vinyl palladium species **3** [14].

Previous work from our group [15,16] demonstrated that this type of cascade can indeed be used to synthesize 7,8 systems (**8**, Figure 1b), and indeed carbopalladation/coupling/electrocyclisation sequences are generally well-established [17,18,19,20]. However, we aimed to avoid the use of a toxic organotin coupling partner as had been employed in our previous work [15,16] by switching to an untested Suzuki/8π strategy. We have also shown that ynamides are viable partners for intramolecular carbopalladation cascades in a carbopalladation/Suzuki/6π sequence in which the ynamide nitrogen atom is located inside the tethering ring (**9**→**10**, Figure 1b) [21]. In our planned work, we now required the nitrogen atom to be positioned at the alkyne terminus, rather than internally. At the outset of this study, several questions therefore had to be answered (Figure 1c):Can carbopalladation/Suzuki/6π cascades proceed with the ynamide nitrogen atom position exocyclic to the forming ring, and how does this alkyne polarity reversal affect reactivity (**11**→**12**, Figure 1c)?Can the bromoenynamide-Suzuki cascade be extended to the synthesis of 8-membered rings (**13**)?Can the resulting 8-membered ring enamides undergo further selective functionalisation towards the carbocyclic D-ring core of lancifodilactone I (**14**)?

## 2. Results and Discussion

### 2.1. Synthesis of 5,6-Fused Systems by Bromoenynamide Carbopalladation/Suzuki Coupling/6π-Electrocyclisation Cascade

To investigate the first of these questions, bromoenynamide **11** was synthesized in four steps from dimethyl malonate **15** (Figure 2). Malonate deprotonation by sodium hydride, followed by allylation with 2,3-dibromopropene, gave the monoallylated product. Further deprotonation by sodium hydride and treatment with excess propargyl bromide gave bromoenyne **16**. Silver catalysed alkyne bromination [22], followed by chemoselective copper catalysed ynamide formation (using the conditions developed by Hsung et al. [23]), gave the desired bromoenynamide **11** in excellent yield.

To study the desired carbopalladation/Suzuki/6π cascade, three vinylboronic acid derivatives (**18**–**20**), and a vinylzinc species (**21**), were investigated as coupling partners. Only the potassium vinyltrifluoroborate salt **20** was found to be effective (Table 1), with other partners resulting only in degradation or no observed reaction. Further optimisation (Table 1) led to identification of tetrakis(triphenylphosphine)palladium(0) and potassium carbonate in THF/water 10:1 at 80 °C as optimal conditions to deliver the desired product **12** in 89% isolated yield.

These results demonstrate that the position of the nitrogen atom on the alkyne (internal vs. external), and therefore the alkyne polarisation, does not influence the carbopalladation and subsequent Suzuki coupling reactivity, with both types of ynamide undergoing the desired cascade in excellent yields [21].

### 2.2. Synthesis of 5,8-Fused Systems by Bromoenynamide Carbopalladation/Suzuki Coupling/8π-Electrocyclisation Cascade

To address the second question, of extending the cascade to formation of 5,8-fused systems from bromoenynamides, a dienylboronic acid equivalent **23** was required (Figure 3). After examination of several different routes, this simple but synthetically challenging fragment was synthesized from 2-methyl-3-butyn-2-ol **22** in two steps. Dehydration [24], followed by rhodium catalysed *trans*-hydroboration developed by Miyaura [25], gave the desired pinacolboronic ester as a mixture of geometric isomers. It was possible to obtain the pure *Z* isomer by preparative HPLC. It is important to note that the stereochemical purity of the diene coupling partner **23** is crucial to the success of the planned cascade: any *E* isomer present in the mixture would preclude an 8π-electrocyclisation, therefore leading to side product formation and yield deterioration. In the event, subjection of this dienyl pinacolboronic ester to the previously optimized conditions cleanly afforded the 5,8-fused product **13** in an excellent yield of 96% (Figure 3).

### 2.3. Selective Functionalisation of 5,6- and 5,8-Fused Ring Systems

To investigate selective oxidations of the 5,6- and 5,8-fused ring scaffolds, hydrolysis of enamide **12** to the corresponding ketone was attempted. Subjection of **12** to a range of strongly or weakly acidic conditions resulted in either no reaction, or decomposition of the starting material (Figure 4a). Attempts to cleave the sulfonamide under common reductive conditions (Mg, MeOH; Na, naphthalene, DME; SmI_2_, pyrrolidine, THF/water) also led to either no reaction, or decomposition of the starting material. To our surprise, *m*-CPBA-mediated epoxidation of **12** selectively epoxidized the tetrasubstituted (non-enamide) alkene (**26**, 59%). Upjohn dihydroxylation of **12** gave a mixture of products, with the major product being **25** (30%). Equivalent transformations of 5,8-fused ring enamide **14** with *m*-CPBA or potassium osmate/NMO resulted in selective oxidation of the trisubstituted alkene (**27**, **28**, Figure 4b), with no reaction observed at the enamide double bond.

These results stand in contrast to our anticipation that the electron donating capabilities of a sulfonamide group, through conjugation of the enamide nitrogen lone pair with its alkene, would render the enamide more nucleophilic and therefore reactive toward oxidation. It was reasoned that conjugation of the nitrogen lone pair with the double bond is reduced or even prevented due to steric congestion in systems **12** and **14**, such that the lone pair may be (near) orthogonal to the alkene π system. With the mesomeric electron-donating effect removed, the inductive electron-withdrawing effects of the sulfonamide group mean the ‘enamide’ is in fact an electron poor olefin, and therefore less reactive toward oxidation.

### 2.4. Attempted Application of the Carbopalladation/Suzuki Coupling/8π-Electrocyclisation Cascade towards the ABCD Rings of Schisandra Triterpenes

To test if the developed cascade could be used in the synthesis of the 7,8- and 7,6- systems found in *Schisandra* triterpenes, model bromoenynamide **34** was synthesized in four steps from dimethyl malonate (Figure 5a). Monoallylated malonate derivative **29** was first subjected to conjugate addition to acrolein, giving aldehyde **30** (47%). Meanwhile, benzyl tosyl amine **31** was transformed to dichloroenamide **32** (73%), which served as a precursor to lithiated ynamide **33** under conditions developed by our group [26]. Treatment of aldehyde **30** with **33** gave a secondary alcohol, which was protected as a *tert*-butyldimethylsilyl ether **34** (30% over two steps).

To our disappointment, under the previously developed conditions using potassium vinyltrifluoroborate, bromoenynamide **34** was converted to a mixture of the ‘direct’ coupling product **38**, and side product **37**, which derives from a formal 7-*endo-trig* Heck-type cyclisation [14] (Figure 5a). Similar reactions of bromoenynamide **34** with diene **23** under a range of conditions resulted only in complex mixtures of unidentifiable products (not shown). These results suggest that in the setting of a 7-membered tether, the vinylpalladium species arising from oxidative addition (**35**) undergoes direct Suzuki coupling at a comparable rate to carbopalladation, and that once carbopalladation has occurred, the resulting dienylpalladium intermediate **36** undergoes intramolecular Heck-type cyclisation to side product **37** much faster than the desired Suzuki coupling.

It was hoped that the more conformationally constrained nature of the ‘real’ substrate (featuring the *Schisandra* AB rings) would reduce the entropic penalty of seven membered ring formation, and hence favour the desired cascade over direct coupling. This substrate **39** was synthesized from aldehyde **6** [27] by addition of the lithiated ynamide derived from **32**. Unfortunately, with only milligram quantities of **39** to hand, attempts to apply the developed carbopalladation, Suzuki coupling, 8π-electrocyclisation cascade resulted only in the formation of complex mixtures of unidentifiable products.

## 3. Materials and Methods

### 3.1. General

All reactions were performed open to air and without precautions to exclude air/moisture unless specified otherwise. Reagents and solvents were purchased from commercial sources and used without further purification unless specified otherwise. NMR spectra were recorded on Bruker AVIII HD 400, NEO 400, AVIII HD 500 and AVII 500 spectrometers. Chemical shifts (δ) are quoted in parts per million (ppm). ^1^H and ^13^C NMR spectra are referenced to residual protons in chloroform-d (δ_H_ = 7.26, δ_C_ = 77.16) and acetone-d_6_ (δ_H_ = 2.05, δ_C_ = 28.95). Peak multiplicities are defined as s (singlet), d (doublet), t (triplet), q (quartet), m (multiplet) and br (broad). Coupling constants (*J*) are reported to the nearest 0.1 Hz. High-resolution mass spectra (HRMS) were recorded on a Thermo Scientific exactive mass spectrometer (Waters Equity autosampler and pump) for electrospray ionisation (ESI). Flash chromatography refers to normal phase column chromatography on silica gel (Merck Si 60, 0.040–0.063 mm) under a positive pressure of nitrogen.

### 3.2. Experimental Procedures

**Dimethyl 2-(3-((*N*-benzyl-4-methylphenyl)sulfonamido)prop-2-yn-1-yl)-2-(2-bromoallyl)-malonate (11).** To a stirred solution of alkyne **16** (50 mg, 0.17 mmol, 1.0 eq.) in acetone (0.34 mL) at room temperature was added AgNO_3_ (2.9 mg, 10 mol%). After stirring for 5 min, *N*-bromosuccinimide (34 mg, 0.19 mmol, 1.1 eq.) was added and the resulting mixture was stirred for a further 4 h at rt. The reaction mixture was concentrated, then pentane was added, and the suspension was filtered through cotton wool to remove the white precipitate. The resulting solution was concentrated to obtain the corresponding bromoalkyne **17** (58 mg, 0.16 mmol), which was of sufficient purity to be used without further purification. Note 1: Because of similar R_f_ values of reactant and product, the reaction was monitored by NMR aliquot (conversion of singlet at 5.84 to 5.80 ppm). Note 2: If acetone is not evaporated completely before trituration with pentane, some succinimide will dissolve and contaminate the product. Note 3: The product was used immediately in the next step due to its tendency to decompose on storage.

To a mixture of *N-*tosylbenzylamine (245 mg, 0.938 mmol, 1.0 eq.), K_3_PO_4_ (398 mg, 1.88 mmol, 2.0 eq.), CuSO_4_ (23 mg, 0.094 mmol, 0.1 eq.) and 1,10-phenanthroline (34 mg, 0.187 mmol, 0.2 eq.) in a vial was added a solution of bromoalkyne **17** (380 mg, 1.03 mmol, 1.1 eq.) in toluene (1 mL). The vial was sealed and heated in an oil bath at 75 °C for 4 days. The reaction mixture was cooled to room temperature, diluted with EtOAc, filtered through Celite and concentrated. The product was purified via flash chromatography (pentane/EtOAc 7:3) to afford **11** (481 mg, 0.877 mmol, 93%) as a pale green gel. **^1^H NMR** (400 MHz, CDCl_3_) δ 7.73 (d, *J* = 8.0 Hz, 2H_Ts_), 7.33 (d, *J* = 8.0 Hz, 2H_Ts_), 7.30–7.27 (m, 3H), 7.25–7.21 (m, 2H), 5.45 (s, 2H), 4.41 (s, 2H), 3.67 (s, 6H), 3.07 (s, 2H), 2.91 (s, 2H), 2.45 (s, 3H). **^13^C NMR** (101 MHz, CDCl_3_) δ 169.8, 144.9, 135.1, 134.8, 130.1, 128.9, 128.9, 128.6, 127.9, 126.3, 123.2, 76.9, 65.6, 56.5, 55.6, 53.8, 53.3, 43.1, 22.0. **HRMS** (ESI) calc. for C_25_H_26_BrNO_6_S [M+H]^+^ 548.0737, found 548.0734.

**Dimethyl 7-((*N*-benzyl-4-methylphenyl)sulfonamido)-1,3,4,5-tetrahydro-2H-indene-2,2-dicarboxylate (12).** A vial loaded with potassium vinyltrifluoroborate (3.7 mg, 0.027 mmol, 1.5 eq.) was taken into the glovebox, and a solution of bromoynamide **11** (10.0 mg, 0.018 mmol, 1.0 eq. in 100 µL) and Pd(PPh_3_)_4_ (2.1 mg, 0.002 mmol, 10 mol% in 300 µL) in previously degassed THF were added. The vial was sealed, removed from the glovebox, and then a degassed solution of K_2_CO_3_ in water (7.6 mg, 0.055 mmol, 3.0 eq. in 50 µL) was quickly added under an inverted cone of nitrogen. The mixture was heated at 80 °C for 10 h. The mixture was cooled to room temperature and directly purified via flash chromatography (pentane/EtOAc 7:3) to afford **12** (8.0 mg, 0.016 mmol, 89%). **^1^H NMR** (400 MHz, acetone) δ 7.74 (d, *J* = 8.1 Hz, 2H), 7.44 (d, *J* = 8.1 Hz, 2H), 7.36–7.15 (m, 5H), 5.20 (t, *J* = 4.6 Hz, 1H), 4.46 (s, 2H), 3.66 (s, 6H), 2.91 (s, 2H), 2.87 (s, 2H), 2.46 (s, 3H), 2.22–2.13 (m, 2H), 2.03 (d, *J* = 9.8 Hz, 2H). ^**13**^**C NMR** (101 MHz, CDCl_3_) δ 172.4, 143.5, 136.2, 136.1, 136.0, 134.8, 131.3, 129.5, 129.3, 128.4, 128.0, 127.8, 125.1, 54.7, 52.9, 43.6, 39.8, 29.8, 23.7, 22.5, 21.7. **HRMS** (ESI) calc. for C_27_H_29_NO_6_S [M+Na]^+^ 518.1608, found 548.1609.

**Dimethyl (6*Z*,8*E*)-9-((*N*-benzyl-4-methylphenyl)sulfonamido)-6-methyl-1,3,4,5-tetra-hydro-2*H*-cyclopenta**[8]**annulene-2,2-dicarboxylate (13).** To a vial loaded with bromoynamide **11** (34 mg, 0.062 mmol, 1.0 eq.) and (*Z*)-boronate ester 23 (18 mg, 0.093 mmol, 1.5 eq.) was added dry and degassed THF (1.3 mL, 50 mM), degassed K_2_CO_3_ solution in water (0.13 mL, 19 mg/mL, 0.19 mmol, 3.0 eq.) and Pd(PPh_3_)_4_ (7.2 mg, 0.006 mmol, 10 mol%). The mixture was further degassed by bubbling N_2_, then sealed and heated overnight at 80 °C. After cooling to rt, water and EtOAc were added, and the organic layer was separated and washed with brine, dried over MgSO_4_ and concentrated. Purification via flash chromatography (pentane/EtOAc 7:3) afforded **13** (32 mg, 0.060 mmol, 96%) of the desired product. **^1^H NMR** (400 MHz, CDCl_3_) δ 7.76 (d, *J* = 8.3 Hz, 2H), 7.37–7.13 (m, 7H), 5.77 (dd, *J* = 5.1, 1.5 Hz, 1H), 5.51 (d, *J* = 4.5 Hz, 1H), 4.55 (s, 2H), 3.71 (s, 6H), 2.91 (s, 2H), 2.61 (s, 2H), 2.44 (s, 3H), 2.02–1.96 (m, 2H), 1.84–1.78 (m, 2H), 1.65 (s, 3H). **^13^C NMR** (101 MHz, CDCl_3_) δ 172.4, 143.4, 143.2, 143.0, 137.8, 136.7, 131.7, 129.6, 129.2, 128.6, 128.5, 128.3, 127.8, 127.6, 120.1, 56.9, 52.9, 52.7, 46.0, 43.4, 30.5, 27.8, 25.4, 21.7. **HRMS** (ESI) calc. for C_30_H_33_NO_6_S [M+H]^+^ 536.2100, found 536.2101.

**Dimethyl 2-(2-bromoallyl)-2-(prop-2-yn-1-yl)malonate (16).** To a stirred suspension of pentane-washed NaH (162 mg pre-wash weight, 4.04 mmol, 1.5 eq., 60% in oil) in dry THF (12 mL) at room temperature under N_2_ was slowly added **29** (676 mg, 2.69 mmol, 1.0 eq.), and the mixture was stirred for 30 min. Propargyl bromide (160 µL, 5.38 mmol, 2.0 eq., 80% wt. in toluene) was slowly added and the reaction mixture was stirred overnight. Diethyl ether and NH_4_Cl (sat., aq.) were added and the organic layer was separated, washed with brine, dried over MgSO_4_ and concentrated. Purification via flash chromatography (pentane/EtOAc 9:1) afforded **16** (490 mg, 1.69 mmol, 63%) as a colourless oil. ^**1**^**H NMR** (400 MHz, CDCl_3_) δ 5.84 (br s, 1H), 5.63 (d, *J* = 1.6 Hz, 1H), 3.77 (s, 6H), 3.31 (br s, 2H), 2.93 (d, *J* = 2.7 Hz, 2H), 2.05 (t, *J* = 2.7 Hz, 1H). Data in agreement with literature values [15].

**(*Z*)-4,4,5,5-Tetramethyl-2-(3-methylbuta-1,3-dien-1-yl)-1,3,2-dioxaborolane (23).** A protocol for *trans*-selective alkyne hydroboration developed by Miyaura and co-workers [25] was applied. To a flame dried Schlenk flask was added anhydrous cyclohexane (27 mL, 0.25 M). The flask was taken into the glovebox and [Rh(cod)Cl]_2_ (66 mg, 0.13 mmol, 1.5 mol%) and *i*-Pr_3_P (86 mg, 0.54 mmol, 6.0 mol%) were added. The flask capped with a septum and removed from the glovebox, then Et_3_N (6.26 mL, 44.9 mmol, 5.0 eq.) and HBPin (1.15 g, 8.99 mmol, 1.0 eq.) were added and the mixture was stirred at room temperature for 30 min. 2-Methyl-1-buten-3-yne** (0.71 g, 10.8 mmol, 1.2 eq.) was added and the mixture was stirred for 3–4 h at room temperature (it could be also left overnight). The reaction was quenched by addition of MeOH (~10 mL) and evaporated (150 mbar, 30 °C). Purification via flash chromatography (pentane/EtOAc 98:2) afforded **23** (850 mg, 5.29 mmol, 49%) as a ~1:1 *Z*/*E* mixture that was separated by reversed phase HPLC (Gemini-NX C18 5 µm, 21×150 mm, H_2_O/MeCN 20:80, 9 mL/min, 6.6 min *Z*-**23**, 7.0 min *E*-**23**) to afford pure *Z* isomer. **^1^H NMR** (500 MHz, CDCl_3_) δ 6.73 (d, *J* = 14.9 Hz, 1H), 5.36 (d, *J* = 14.9 Hz, 1H), 5.07–5.04 (m, 2H), 1.96 (s, 3H), 1.29 (s, 12H). **^13^C NMR** (126 MHz, CDCl_3_) δ 148.6, 144.0, 119.2, 83.7, 24.9, 20.5. The alkene carbon atom bonded to boron was not observed due to quadrupolar relaxation. *E* isomer **^1^H NMR** (400 MHz, CDCl_3_) δ 7.11 (d, *J* = 18.2 Hz, 1H), 5.56 (d, *J* = 18.2 Hz, 1H), 5.16 (s, 2H), 1.85 (s, 3H), 1.29 (s, 12H). **2-Methyl-1-buten-3-yne (b.p. 35 °C) could be synthesized from 2-methylbut-3-yn-2-ol according to a literature procedure [24], or purchased directly from a commercial source. **Dimethyl (3a*R*,7a*R*)-7-((*N*-benzyl-4-methylphenyl)sulfonamido)-3a,7a-dihydroxy-1,3,3a,4,5,7a-hexahydro-2*H*-indene-2,2-dicarboxylate (25).** To a solution of diene **12** (10.0 mg, 0.020 mmol) in acetone (0.30 mL) was added NMO·H_2_O (6.2 mg, 0.040 mmol, 2.0 eq.) and solution of K_2_OsO_2_(OH)_4_ (0.7 mg, 0.002 mmol, 10 mol%) in water (0.10 mL). The reaction mixture was stirred for 4 h at room temperature. Aqueous Na_2_S_2_O_3_ and EtOAc were added, and the organic layer was washed with sat. NaHCO_3_, brine, dried (MgSO_4_) and concentrated. Purification via flash chromatography (pentane/EtOAc 1:1) afforded **25** (3.2 mg, 0.006 mmol, 30%). ^**1**^**H NMR** (400 MHz, CDCl_3_) δ 7.69 (d, *J* = 8.3 Hz, 1H), 7.33–7.19 (m, 2H+5H), 5.27 (dd, *J* = 5.0, 3.0 Hz, 1H), 4.57 (d, *J* = 14.1 Hz, 1H), 4.41 (d, *J* = 14.1 Hz, 1H), 3.74 (s, 3H), 3.63 (s, 3H), 2.64 (d, *J* = 14.4 Hz, 1H), 2.50–2.44 (m, 1H), 2.44 (s, 3H), 2.31–2.19 (m, 1H +1H), 2.06–1.97 (m, 1H) 1.95–1.86 (m, 1H), 1.82–1.74 (m, 1H), 1.72–1.60 (m, 1H). **^13^C NMR** (126 MHz, CDCl_3_) δ 172.8, 144.2, 137.5, 135.7, 135.4, 134.1, 129.9, 129.6, 128.6, 128.6, 128.3, 79.5, 56.7, 55.7, 53.2, 53.1, 44.6, 44.5, 30.5, 22.7, 21.7. **HRMS** (ESI) calc. for C_27_H_31_NO_8_S [M+Na]^+^ 552.1663, found 552.1659.

**Dimethyl (3a*R*,7a*R*)-7-((*N*-benzyl-4-methylphenyl)sulfonamido)-4,5-dihydro-1*H*-3a,7a-epoxyindene-2,2(3*H*)-dicarboxylate (26).** To a solution of diene **12** (10.0 mg, 0.020 mmol) in CH_2_Cl_2_ (0.40 mL) was added NaHCO_3_ (2.0 mg, 0.024 mmol, 1.2 eq.) and *m*-CPBA (70%, 5.5 mg, 0.022 mmol, 1.1 eq.) and the mixture was stirred for 30 min at rt. The colour turned from red to yellow upon addition of *m*-CPBA. The reaction was diluted with CH_2_Cl_2_, and the organic layer was washed with aqueous NaHCO_3_, Na_2_S_2_O_3_ and brine, dried (MgSO_4_) and concentrated. Purification via flash chromatography (pentane/EtOAc 7:3 to 1:1) afforded **26** (6.1 mg, 0.012 mmol, 59%) of the epoxide product. ^**1**^**H NMR** (400 MHz, CDCl_3_) δ 7.70 (d, *J* = 8.3 Hz, 1H), 7.35–7.21 (m, 2H+5H), 5.29 (t, *J* = 4.9 Hz, 1H), 4.65 (d, *J* = 13.5 Hz, 1H), 4.26 (d, *J* = 13.5 Hz, 1H), 3.70 (s, 3H), 3.66 (s, 3H), 3.02 (d, *J* = 14.2 Hz, 1H), 2.96 (d, *J* = 14.2 Hz, 1H), 2.44 (s, 3H), 2.19–1.96 (m, 5H), 1.48 (dt, *J* = 14.0, 9.1 Hz, 1H). **^13^C NMR** (126 MHz, CDCl_3_) δ 171.4, 171.3, 143.8, 135.8, 135.6, 134.8, 131.1, 129.8, 129.6, 128.5, 128.1, 128.1, 68.8, 63.5, 56.6, 55.2, 53.1, 53.0, 38.9, 36.3, 22.1, 21.9, 21.7. **HRMS** (ESI) calc. for C_27_H_29_NO_7_S [M+Na]^+^ 534.1557, found 534.1550.

**Dimethyl (6*R*,7*S*,*E*)-9-((*N*-benzyl-4-methylphenyl)sulfonamido)-6,7-dihydroxy-6-methyl-1,3,4,5,6,7-hexahydro-2*H*-cyclopenta[8]annulene-2,2-dicarboxylate (27).** To a solution of triene **14** (10.0 mg, 0.019 mmol) in acetone (0.30 mL) was added NMO.H_2_O (3.4 mg, 0.040 mmol, 2.0 eq.) and a solution of K_2_OsO_2_(OH)_4_ (0.7 mg, 0.002 mmol, 10 mol%) in water (0.10 mL). The reaction mixture was stirred at rt, 3 h. Aqueous Na_2_S_2_O_3_ and EtOAc were added, and the organic layer was washed with sat. NaHCO_3_, brine, dried (MgSO_4_) and concentrated. Purification via flash chromatography (pentane/EtOAc 1:1) afforded **27** (7.5 mg, 0.013 mmol, 71%). ^**1**^**H NMR** (400 MHz, CDCl_3_) δ 7.77 (d, *J* = 8.2 Hz, 1H_Ts_), 7.35–7.14 (m, 2H+5H), 5.41 (d, *J* = 7.6 Hz, 1H), 4.95 (d, *J* = 14.4 Hz, 1H), 4.15 (d, *J* = 14.4 Hz, 1H), 3.93 (d, *J* = 7.6 Hz, 1H), 3.75 (s, 3H), 3.67 (s, 3H), 3.01–2.91 (m, +1H), 2.76 (d, *J* = 17.1 Hz, 1H), 2.44 (s, 3H), 2.39–2.31 (m, 1H), 2.07–1.90 (m, 2H), 1.25–1.20 (m, 2H), 1.19 (s, 3H). **^13^C NMR** (126 MHz, CDCl_3_) δ 172.8, 144.2, 137.5, 135.7, 135.4, 134.1, 129.9, 129.6, 128.6, 128.6, 128.3, 79.5, 56.7, 55.7, 53.2, 53.1, 44.6, 44.5, 30.5, 22.7, 21.7. **HRMS** (ESI) calc. for C_27_H_31_NO_8_S [M+Na]^+^ 552.1663, found 552.1659.

**Dimethyl (1a*R*,8a*S*,*E*)-7-((*N*-benzyl-4-methylphenyl)sulfonamido)-1a-methyl-1a,2, 3,4,6,8a-hexahydro-5*H*-cyclopenta[5,6]cycloocta[1,2-*b*]oxirene-5,5-dicarboxylate (28).** To a solution of triene **14** (20 mg, 0.037 mmol) in CH_2_Cl_2_ (0.5 mL) was added NaHCO_3_ (5 mg, 0.056 mmol, 1.5 eq.) and *m*-CPBA (70%, 10 mg, 0.041 mmol, 1.1 eq.). The mixture was stirred overnight at rt, diluted with CH_2_Cl_2,_ and washed with aqueous Na_2_S_2_O_3_ (aq., sat.), NaHCO_3_ (aq., sat.), and brine, dried (MgSO_4_) and concentrated. Purification via flash chromatography (pentane/EtOAc 7:3) afforded **28** (8.7 mg, 0.016 mmol, 42%). **^1^H NMR** (400 MHz, CDCl_3_) δ 7.76 (d, *J* = 8.2 Hz, 2H), 7.32 (d, *J* = 8.2 Hz, 2H), 7.25–7.18 (m, 3H), 7.14–7.07 (m, 2H), 5.54 (s, 1H), 4.86 (d, *J* = 15.0 Hz, 1H), 4.18 (d, *J* = 15.0 Hz, 1H), 3.76 (s, 3H), 3.70 (s, 3H), 3.24 (s, 1H), 2.98–2.84 (m, 2H), 2.81 (d, *J* = 16.0 Hz, 1H), 2.56 (d, *J* = 16.0 Hz, 1H), 2.45 (s, 3H), 1.89–1.84 (m, 2H), 1.49 (dt, *J* = 13.8, 4.4 Hz, 1H), 1.13 (s, 3H), 1.06–0.96 (m, 1H). **^13^C NMR** (101 MHz, CDCl_3_) δ 172.3, 172.1, 143.9, 137.6, 136.3, 129.8, 128.7, 128.5, 127.9, 127.8, 127.7, 124.8, 61.0, 59.9, 57.2, 53.1, 53.0, 51.8, 47.2, 43.8, 29.8, 27.3, 21.7. **HRMS** (ESI) calc. for C_30_H_33_NO_7_S [M+H]^+^ 552.2050, found 552.2052.

**Dimethyl 2-(2-bromoallyl)malonate (29).** To a stirred suspension of pentane-washed NaH (1.09 g pre-wash weight, 27.2 mmol, 1.2 eq., 60% in oil) in dry THF (68 mL) at room temperature under N_2_ was slowly added dimethyl malonate (2.61 mL, 22.7 mmol, 1.0 eq.), and the mixture was stirred for 30 min. 2,3-Dibromopropene (2.22 mL, 22.7 mmol, 1.0 eq. neat) was slowly added and the reaction mixture was stirred overnight. Diethyl ether and NH_4_Cl (sat., aq.) were added. The organic layer was separated, washed with brine, dried over MgSO_4_ and concentrated. The product was distilled under reduced pressure (0.5 mbar) and three fractions were collected. The first fraction (45 °C) contained unreacted dimethyl malonate, the second (65–90 °C) contained pure monoallylated product and the third fraction (100–110 °C) contained mostly diallylated product (these products were very hard to separate by chromatography). The desired product **29** was thus obtained (3.49 g, 13.9 mmol, 61%) as a colourless liquid. **^1^H NMR** (400 MHz, CDCl_3_) δ 5.69 (s, 1H), 5.47 (s, 1H_9_), 3.82 (t, *J* = 7.5 Hz, 1H), 3.75 (s, 6H), 3.02 (d, *J* = 7.5 Hz, 2H). Data in agreement with literature values [28].

**Dimethyl 2-(2-bromoallyl)-2-(3-oxopropyl)malonate (30).** A modified literature protocol for acrolein conjugate addition was applied [29]. To a mixture of acrolein (120 µL, 1.78 mmol, 1.0 eq.) and bromoallyl malonate **29** (448 mg, 1.78 mmol, 1.0 eq.) in MeOH (5.2 mL) at room temperature was slowly added NaOMe (25% wt solution in MeOH, 82 µL, 0.36 mmol, 0.2 eq.). The resulting mixture was stirred for 4 h, then the solvent was evaporated under reduced pressure. The residue was dissolved in Et_2_O and washed with water, brine, dried over MgSO_4_, filtered and concentrated under reduced pressure. The residue was purified via flash column chromatography (pentane/EtOAc 7:3) to give **30** (259 mg, 0.84 mmol, 47%) as a viscous liquid. ^**1**^**H NMR** (400 MHz, CDCl_3_) δ 9.73 (t, *J* = 1.3 Hz, 1H), 5.67 (dt, *J* = 1.8, 0.9 Hz, 1H), 5.60 (d, *J* = 1.8 Hz, 1H), 3.74 (s, 6H), 3.18 (d, *J* = 0.9 Hz, 2H), 2.52–2.46 (m, 2H), 2.37–2.27 (m, 2H). ^**13**^**C NMR** (101 MHz, CDCl_3_) δ 200.3, 170.5, 126.6, 122.3, 56.3, 52.8, 44.0, 39.1, 24.4.

**(*E*)-*N*-Benzyl-*N*-(1,2-dichlorovinyl)-4-methylbenzenesulfonamide (32).** Prepared according to a literature procedure [26]. To a stirred suspension of *N-*tosylbenzylamine (1.50 g, 5.74 mmol, 1.0 eq.) and powdered Cs_2_CO_3_ (2.81 g, 8.61 mmol, 1.5 eq.) in DMF (4.3 mL), at 50 °C was added trichloroethylene (0.57 mL, 6.31 mmol, 1.1 eq.) dropwise over 10 min. The resulting mixture was stirred at 50 °C until the reaction reached completion as judged by TLC (~1–2 h). The mixture was cooled to room temperature, then partitioned between EtOAc and H_2_O. The organic layer was separated and further washed with water (×2) and brine. The organic layer was then dried (MgSO_4_), filtered and concentrated. Recrystallisation from EtOAc (10–20 mL) afforded **32** (1.50 g, 4.20 mmol, 73%) as white crystals. ^**1**^**H NMR** (400 MHz, CDCl_3_) δ 7.86 (d, *J* = 8.3 Hz, 2H), 7.36 (d, *J* = 8.3 Hz, 2H), 7.34–7.27 (m, 5H), 6.27 (s, 1H), 5.08–3.69 (very br s, 2H) 2.47 (s, 3H). ^**13**^**C NMR** (101 MHz, CDCl_3_) δ 144.7, 135.2, 133.4, 129.8, 129.4, 128.5, 128.5, 128.4, 121.7, 77.2, 51.8, 21.7. Data in agreement with literature values [26].

**Dimethyl 2-(5-((*N*-benzyl-4-methylphenyl)sulfonamido)-3-((*tert*-butyldimethylsilyl)oxy)-pent-4-yn-1-yl)-2-(2-bromoallyl)malonate (34).** Synthesized according to the protocol for ynamide synthesis developed by Anderson et al. [26]. To an oven dried, argon flushed flask was added 1,2-dichloroenamide **32** (116 mg, 0.33 mmol, 2.5 eq.) and anhydrous THF (1.1 mL), and the mixture was cooled to −78 °C whilst stirring. A solution of phenyllithium (1.9 M solution in dibutyl ether, 0.34 mL, 5.0 eq.) was then added dropwise, and the mixture was left to stir at −78 °C for 15 min. After complete conversion to the lithiated ynamide (as confirmed by TLC consumption of the dichloroenamide), a solution of the aldehyde **30** (40 mg, 0.13 mmol, 1.0 eq.) in anhydrous THF (0.6 mL) was added at −78 °C and the mixture was stirred for 1 h. The reaction was quenched with NH_4_Cl (at −78 °C, sat., aq.), then warmed to room temperature and extracted with Et_2_O. The organic extract was washed with brine, dried (MgSO_4_) and concentrated. The crude product was used directly in the next step.

To a stirred solution of the crude alcohol in DMF (1 mL) at room temperature was added imidazole (14 mg, 0.19 mmol, 1.5 eq.) and TBSCl (20 mg, 0.13 mmol, 1.0 eq.), and the mixture was stirred for 1 h until complete (as monitored by TLC). Water and Et_2_O were added, then the organic layer was separated and washed with brine, dried (MgSO_4_) and concentrated. Since the ynamide impurity (resulting from protonation of excess lithiated ynamide from previous step) has a similar R_f_ value to the product, it was removed by washing the crude product with pentane several times (until the yellow colour faded). A solid, white impurity that is not soluble in pentane remains as a precipitate. The pentane washes were concentrated, and the residue was purified via flash chromatography (pentane/EtOAc 85:15) to afforded **34** (28 mg, 0.04 mmol, 30% over two steps) as an oil. **^1^H NMR** (400 MHz, d_6_-acetone) δ 7.84 (d, *J* = 8.3 Hz, 2H), 7.49 (d, *J* = 8.3 Hz, 2H), 7.40–7.24 (m, 5H), 5.81–5.69 (m, 1H), 5.56 (d, *J* = 1. Hz, 1H), 4.54 (s, 1H), 4.52 (s, 1H), 4.50–4.46 (m, 1H), 3.72 (s, 3H), 3.71 (s, 3H), 3.13 (s, 2H), 2.48 (s, 3H), 2.15–2.09 (m, 2H), 1.54–1.42 (m, 2H), 0.83 (s, 9H), 0.01 (s, 3H), −0.04 (s, 3H). ^**13**^**C NMR** (101 MHz, d_6_-acetone) δ 170.4, 144.9, 135.1, 134.9, 130.0, 128.7, 128.5, 128.2, 127.7, 126.9, 122.1, 78.2, 72.0, 62.5, 56.4, 55.4, 52.1, 52.1, 42.9, 33.3, 27.2, 25.3, 20.7, 17.8, −5.3, −5.9. **HRMS** (ESI) calc. for C_33_H_44_BrNO_7_SSi [M+Na]^+^ 728.1683, found 728.1679.

**Dimethyl 8-((*N*-benzyl-4-methylphenyl)sulfonamido)-6-((*tert*-butyldimethylslyl)-oxy)bicyclo[5.2.0]nona-1,7-diene-3,3-dicarboxylate (37)** and **dimethyl 2-(5-((*N*-benzyl-4-methylphenyl)sulfonamido)-3-((*tert*-butyldimethylsilyl)oxy)pent-4-yn-1-yl)-2-(2-methylenebut-3-en-1-yl)malonate (38).** To a vial loaded with bromoynamide **34** (11 mg, 0.016 mmol, 1.0 eq.) and potassium vinyltrifluoroborate **20** (3.1 mg, 0.023 mmol, 1.5 eq.) was added dry and degassed THF (0.40 mL), degassed K_2_CO_3_ solution in water (6.4 mg in 40 µL water, 0.047 mmol, 3.0 eq.) and Pd(PPh_3_)_4_ (1.8 mg, 0.002 mmol, 10 mol%). The mixture was further degassed by bubbling N_2_, then sealed and heated overnight at 80 °C. After cooling to rt, water and EtOAc were added, the organic layer was separated and washed with brine, dried over MgSO_4_ and concentrated. Purification via flash chromatography (pentane/EtOAc 7:3) afforded undesired products **37** (2.8 mg, 0.005 mmol, 28%) and 38 (3.1 mg, 0.005 mmol, 30%). 37: **^1^H NMR** (400 MHz, Acetone) δ 7.76 (d, *J* = 8.4 Hz, 2H), 7.52–7.38 (m, 2H), 7.40–7.24 (m, 5H), 5.14 (d, *J* = 16.8 Hz, 1H), 5.00 (s, 1H), 4.79 (d, *J* = 16.8 Hz, 1H), 4.28 (br s, 1H), 3.68 (s, 3H), 3.65 (s, 3H), 3.23–3.18 (m, 2H), 2.44 (s, 3H), 2.44–2.40 (m, 1H), 2.08–2.02 (m, 1H), 1.82–1.71 (m, 1H), 1.68–1.58 (m, 1H), 0.79 (s, 9H), −0.00 (s, 3H), −0.02 (s, 3H). 38: **^1^H NMR** (400 MHz, CDCl_3_) δ 7.74 (d, *J* = 8.3 Hz, 2H), 7.34–7.27 (m, 5H+2H), 6.25 (dd, *J* = 17.5, 10.9 Hz, 1H), 5.20 (d, *J* = 17.5 Hz, 1H), 5.14 (s, 1H), 5.00 (d, *J* = 10.9 Hz, 1H), 4.94 (s, 1H), 4.51 (d, *J* = 14.0 Hz, 1H), 4.43 (d, *J* = 14.0 Hz, 1H), 4.33 (t, *J* = 6.1 Hz, 1H), 3.68 (s, 3H), 3.67 (s, 3H), 2.82 (s, 2H), 2.44 (s, 3H), 2.07–1.85 (m, 2H), 1.54–1.39 (m, 2H), 0.81 (s, 9H), −0.04 (s, 3H), −0.08 (s, 3H). Further structural assignment is provided in the Appendix A.

**N-Benzyl-N-(4-((3*S*,3a*R*,6a*R*)-3a-(2-bromoallyl)-2,2-dimethyl-5-oxohexahydrofuro[3,2-*b*]furan-3-yl)-3-hydroxybut-1-yn-1-yl)-4-methylbenzenesulfonamide (39).** To an oven dried, argon flushed flask was added 1,2-dichloroenamide **32** (18 mg, 0.051 mmol, 3.0 eq.) and anhydrous THF (0.5 mL), and the mixture was cooled to −78 °C. A solution of phenyllithium (1.9 M solution in dibutyl ether, 25 µL, 2.8 eq.) was then added dropwise, and the mixture was left to stir at −78 °C for 15 min. After almost complete conversion to the lithiated ynamide (the 1,2-dichloroenamide is in excess, so a small amount remains), the aldehyde 6 (5.4 mg, 0.017 mmol) in anhydrous THF (0.5 mL) was added at −78 °C and stirred for 1 h. The reaction was quenched by the addition of NH_4_Cl (at −78 °C, sat., aq.), then warmed to room temperature and extracted with Et_2_O. The organic extract was washed with brine, dried (MgSO_4_) and concentrated. The product was purified via flash chromatography to afford **39** (5.1 mg, 0.008 mmol, 50%) as a 2:1 mixture of diastereomeric alcohols. **^1^H NMR** (500 MHz, CDCl_3_) δ 7.78–7.74 (m, 2H), 7.39–7.25 (m, 2H+5H), 5.77–5.63 (m, 2H), 4.83–4.79 (m, 1H), 4.58–4.54 (m, 1H), 4.51–4.43 (m, 2H), 3.37–3.25 (m, 1H), 2.93–2.82 (m, 1H), 2.71–2.59 (m, 1H), 2.59–2.52 (m, 1H), 2.46 (s, 3H), 2.29 (dd, *J* = 9.6, 3.9 Hz, 1H), 1.94 (d, *J* = 5.6 Hz, 1H), 1.74–1.67 (m, 1H), 1.46–1.38 (m, 1H), 1.13 (s, 3H), 1.02 (s, 3H). **HRMS** (ESI) calc. for C_29_H_32_BrNO_6_S [M+H]^+^ 602.1207, found 602.1207. Due to insufficient material, the product structure was assigned using ^1^H NMR, ^1^H–^1^H COSY and ^13^C edited-HSQC (see the Appendix A).

## 4. Conclusions

In conclusion, a tandem carbopalladation, Suzuki coupling and 6π- or 8π-electrocyclisation reaction of bromoenynamides in which the ynamide nitrogen is exocyclic to the forming ring was explored in the context of synthesis of terpenes from the *Schisandraceae* family. It was shown that this tactic can be applied to the synthesis of 5,6- and 5,8-fused bicyclic ring systems in high yields. A route to a versatile dienylboronic ester building block was also developed. The reactivity of the resulting enamides was explored, and it was found that reactivity patterns differ to those observed with ynamides. Synthesis of 7,5- and 7,8 ring systems using the developed cascade was tested; however, the greater difficulty of 7-membered ring closure and propensity to form 7,4-fused ring by-products hindered the application of this cascade. Nevertheless, for smaller ring tethers, the tandem carbopalladation, Suzuki coupling and 6/8π electrocyclisation was shown to be a highly efficient tactic, allowing a rapid increase in molecular complexity, and with a novel positioning of the ynamide nitrogen group. This process may therefore enable future applications in the synthesis of other classes of polycyclic natural products.

## Figures and Tables

**Figure 1 molecules-28-04468-f001:**
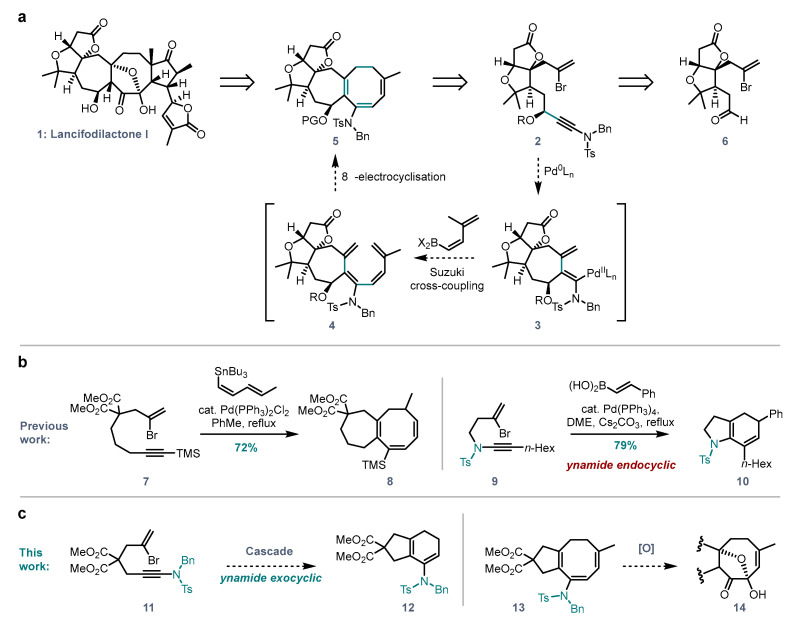
(**a**) Retrosynthetic strategy towards lancifodilactone I. (**b**) Synthesis of 7,8-fused systems via alkyne carbopalladation/Stille coupling/8π-electrocyclisation tandem sequence, and synthesis of 5,6-fused systems via ynamide carboplladation/Suzuki coupling/6π-eletrocyclisation sequence. (**c**) Proposed synthesis of 5,6- or 5,8-fused ring systems with external nitrogen position, and subsequent selective oxidations.

**Figure 2 molecules-28-04468-f002:**
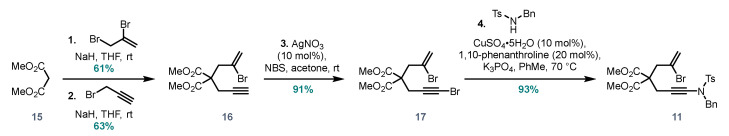
Synthesis of bromoynamide **11**.

**Figure 3 molecules-28-04468-f003:**
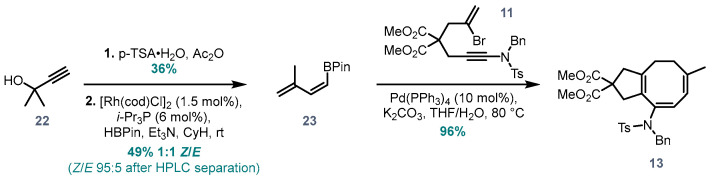
Synthesis of (*Z*)-3-methylbutadienyl boronate ester **23** and application of the optimized cascade to synthesis of 5,8-fused system **13**.

**Figure 4 molecules-28-04468-f004:**
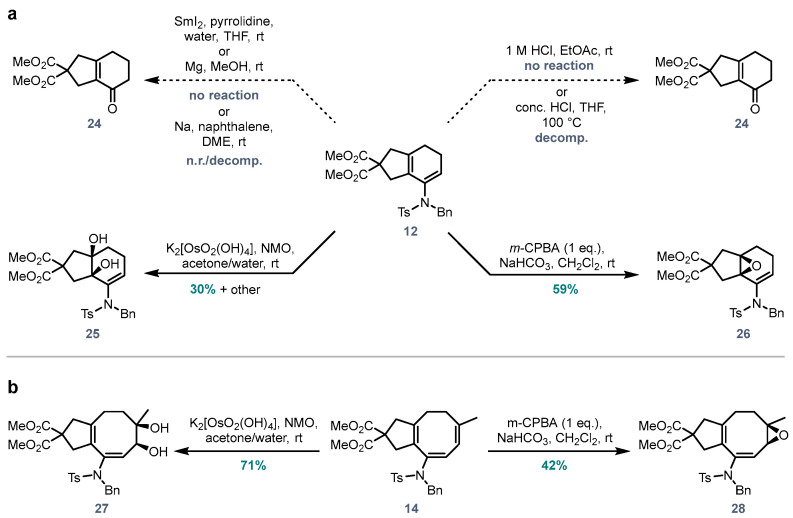
(**a**) Oxidations of the 5,6-fused ring system **12**; (**b**) Oxidations of the 5,8-fused ring system **14**.

**Figure 5 molecules-28-04468-f005:**
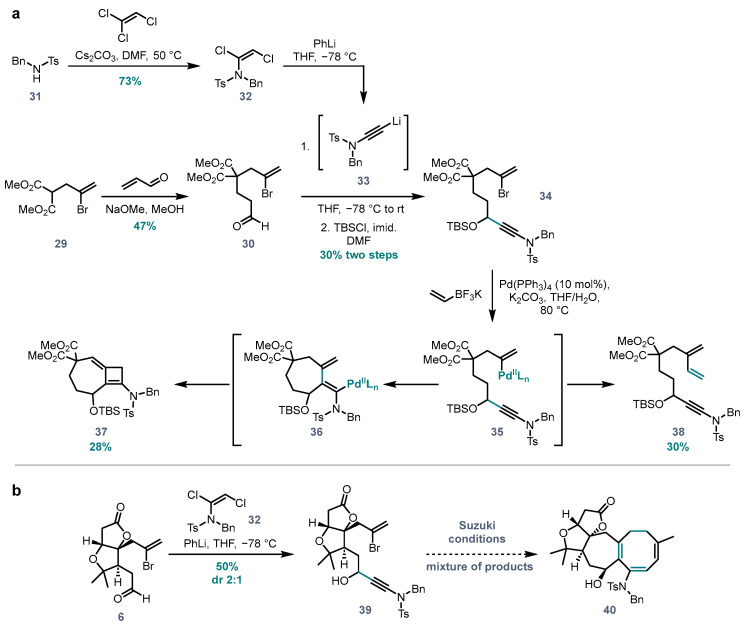
(**a**) Application of the cascade to the synthesis of 7,6-fused system; (**b**) application of the cascade to the “real” substrate **39**.

**Table 1 molecules-28-04468-t001:** Optimisation of the carbopalladation/Suzuki coupling/6π-electrocyclisation cascade.

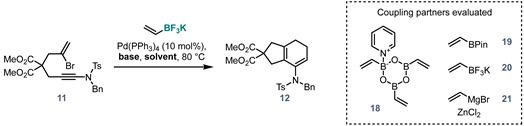			
	BASE		
Solvent	K_2_CO_3 (aq)_	Cs_2_CO_3 (s)_	LiOH _(aq)_
Toluene	0 (100)	0 (24)	38 (80)
THF	100 (100)	7 (100)	25 (66)
Dioxane	88 (100)	34 (100)	27 (73)
THF/H2O (10:1)	89 ^a^	–	–

Yield (%) and conversion (%, indicated in parentheses) for the reaction of **11** with potassium vinyltrifluoroborate salt **20**, determined by HPLC analysis after 10 h at 80 °C. ^a^ Isolated yield. _aq_—aqueous solution, _s_—solid.

## Data Availability

Copies of ^1^H and ^13^C NMR spectra, as well as 2D NMR spectra and associated discussion and structural assignments, are provided in the Appendix A.

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
