# Peer review of "Synthetic Study toward Triterpenes from the Schisandraceae Family of Natural Products"

_molecules, 2023, doi:10.3390/molecules28114468_

Round 1

Reviewer 1 Report

The paper describes the methodology to the congested carbon framework in the polycyclic triterpenes, featuring a unique carbopalladation/Suzuki coupling/6π or 8π electrocyclic reaction cascade. The novel feature is the incorporation of nitrogen substituent at the alkyne terminal, which may be useful for functionalization of the molecule at later stages. However, authors’ plan aborted due to less reactivity of the enamide functional group after the formation of the intended bicyclic ring systems (Figure 4). Another disappointment is inaccessibility to expected 7,8- and 7,6-ring systems, probably because of the flexibility of the molecule undesired for the seven-membered ring formation (Figure 5). Nevertheless, this short-cut strategy to bicyclic carbon frameworks (found in many bioactive terpenoids) from the acyclic synthetic precursors deserves publication. In whole, the manuscript is logical and well written, and therefore, the reviewer recommends publication of this paper after minor revision.

There are two concerns.

In line 84, authors mentioned that “Only the potassium vinyltrifluoroborate salt 20 was found to be effective (Table 1), with other partners resulting only in degradation or no observed reaction.” However, authors selected a BPin derivative 23, rather than trfluoroborate, as a primal coupling partner [with mentioning that “a dienylboronic acid equivalent 23 was required (Figure 3)” in line 102], and actually they obtained the product 13 in 95%(!) yield with the same reaction conditions as in Table 1. The reviewer’s concern is that if the coupling of 11 and 23 proceeds in a very high yield, the reaction between 11 and 19 should give some products (12 or ‘direct’ coupling product, or others). Consequently, the reviewer suspects the accuracy of the comment on line 84. Is it impossible to prepare the trifluoroborate equivalent of 23?

In Figure 5a, authors depicted two unexpected products, ‘direct’ Suzuki coupling product 38 and unusual bicyclo[5.2.0]nonadiene (7,4-ring product) 37, around 30% yield each. However, no spectroscopic evidence is provided even in experimental section and Supporting Information. Especially the latter compound is rare, so they need evidence for it.

Other suggestions are listed in below.

10: synthesized (correct all)

14: ring system

15: syntheses

16: represents the first cyclisations

Figure 2: Check misprints on ‘dot’ and ‘degree’. (check all)

146: bromoenynamide 34 (?)

4.1 Experimental procedures:

1) There are many discrepancies between the yields in the Schemes and the experimental section.

2) The order of experiments should match the order of Schemes and Tables in the text. (For instance, the syntheses of 16 and 13).

3) Experiment from 34 to 37/38 should be included.

243-253: The synthesis of 16 should start from 15. Compound 29 in line 245 is incorrect.

254-272: The synthesis of 23 should start from 22. This experimental procedure should place after line 227. In addition, it is recommended that the data of E-isomer is added (at least, 1H NMR spectrum). Such attention increases the quality of the paper.

294: 0.012 mmol(?)

304: 2 eq(?) (not 1.2 eq)

Reviewer 2 Report

Kravljanac and Anderson report an interesting palladium-catalyzed cascade reaction of bromovinyl-tethered ynamides and vinylboron reagents to form 5/8 membered ring systems, which are the basis of several terpenoid natural products. In case of the desired 7/8 ring system, the cascade stalled however and either an undesired side reaction occurred or the product could not be isolated from a complex mixture.

.While the use of alkynylsilanes as coupling components has already been demonstrated by the same authors, the innovative aspect of the present work is the extension to ynamides, which was only a partial success. To arrive at the natural products, the enamide moiety resulting from the ynamide function should be oxidatively functionalized but the expected selectivity over the other olefinic double bonds could not be achieved and the nitrogen remained in the product.

While the substrate scope of the cascade reaction could be expanded to ynamides for certain types of substrates, the outcome was not beneficial in view of the planned synthetic strategy so that the publication largely reports a failed attempt. Nevertheless, there is some value to report such failures and what has been working instead of the desired route could be used as the basis for further developments.

The manuscript is written nicely and my only correction would be the symbol used for copper sulfate pentahydrate in Figure 2. The SI is of good quality.

Reviewer 3 Report

Anderson and Kravljanac reported the palladium-catalysed cascade cyclisation of a bromoenynamide, Suzuki coupling and 6/8π-electrocyclisation towards the formation of bicyclic enamides. Although the final target was not achieved for the synthesis of 7-membered ring closure, which leading to side product, the developed system still present a highly efficient tactic for the formation of bicyclic enamides, which may find applications in other synthetic contexts.

Figure 5 Compound 34 was wrong.

The purity of some of the products, such as 12, need to be improved further.

OK
